# Gradient Descent Meets Shift-and-Invert Preconditioning for Eigenvector Computation

**Zhiqiang Xu**
Cognitive Computing Lab (CCL), Baidu Research
National Engineering Laboratory of Deep Learning Technology and Application, China
xuzhiqiang04@baidu.com

## Abstract

Shift-and-invert preconditioning, as a classic acceleration technique for the leading eigenvector computation, has received much attention again recently, owing to fast least-squares solvers for efficiently approximating matrix inversions in power iterations. In this work, we adopt an inexact Riemannian gradient descent perspective to investigate this technique on the effect of the step-size scheme. The shift-and-inverted power method is included as a special case with adaptive step-sizes. Particularly, two other step-size settings, i.e., constant step-sizes and Barzilai-Borwein (BB) step-sizes, are examined theoretically and/or empirically. We present a novel convergence analysis for the constant step-size setting that achieves a rate at $\tilde{O}(\sqrt{\frac{\lambda_1}{\lambda_1 - \lambda_{p+1}}})$, where $\lambda_i$ represents the $i$-th largest eigenvalue of the given real symmetric matrix and $p$ is the multiplicity of $\lambda_1$. Our experimental studies show that the proposed algorithm can be significantly faster than the shift-and-inverted power method in practice.

## 1 Introduction

Eigenvector computation is a fundamental problem in numerical algebra and often of central importance to a variety of scientific and engineering computing tasks such as principal component analysis [Fan et al., 2018], spectral clustering [Ng et al., 2001], low-rank matrix approximation [Hastie et al., 2015, Liu and Li, 2014], among others. Classic solvers for this problem are power methods and Lanczos algorithms [Golub and Van Loan, 1996]. Although Lanczos algorithms possess the optimal convergence rate $\tilde{O}(\frac{1}{\sqrt{\lambda_1 - \lambda_2}})$, it seems not amenable to stochastic optimization. People thus tend to develop faster algorithms on top of power methods [Arora et al., 2012, 2013, Hardt and Price, 2014, Shamir, 2015, Garber and Hazan, 2015, Garber et al., 2016, Lei et al., 2016, Wang et al., 2017]. One notable technique among them is the shift-and-invert preconditioning that has revived recently for this purpose [Garber and Hazan, 2015, Garber et al., 2016, Wang et al., 2017, Gao et al., 2017]. Using this technique, each power iteration step can be reduced to approximately solving a linear system subproblem that can leverage fast least-squares solvers, e.g., accelerated gradient descent (AGD) [Nesterov, 2014] or stochastic variance reduced gradient (SVRG) [Johnson and Zhang, 2013].

In this work, we take a Riemannian gradient descent view to investigate the shift-and-invert preconditioning for the leading eigenvector computation on the effect of the step-size scheme. The resulting algorithm thus is termed as the shift-and-inverted Riemannian gradient descent eigensolver, or SI-rgEIGS for short. It includes the shift-and-invert preconditioned power method (termed as SI-PM for short) as a special case with adaptive step-sizes. Applying the shift-and-invert preconditioning technique needs to locate an appropriate upper bound of the largest eigenvalue, i.e., $\sigma > \lambda_1$, as the shift parameter. We reply on the crude phase of the shift-and-inverted power method [Garber and Hazan, 2015, Garber et al., 2016] to get this upper bound in theory. However, in practice, the plain power

method often works via the proposed heuristics in experiments. In addition, the crude phase can warm-start the Riemannian gradient descent method. Similarly, Shamir [2016a] adopted the plain power method to warm-start the stochastic variance reduced projected gradient descent without pre-conditioning for principal component analysis (VR-PCA). The crude phase only consumes non-dominant time due to the independence of the final accuracy parameter $\epsilon$ [Wang et al., 2017]. The algorithm then steps into an accurate phase by calling the Riemannian gradient descent solver on the shift-and-inverted matrix $(\sigma \mathbf{I} - \mathbf{A})^{-1}$, i.e., solving the following problem:

$$\min_{\mathbf{x} \in \mathbb{R}^{n \times 1} : \|\mathbf{x}\|_2 = 1} h(\mathbf{x}) = -\frac{1}{2}\mathbf{x}^\top (\sigma \mathbf{I} - \mathbf{A})^{-1} \mathbf{x}. \tag{1}$$

In each gradient descent step, we have to solve a linear system $(\sigma \mathbf{I} - \mathbf{A})\mathbf{z} = \mathbf{x}_{t-1}$ in order to get the Euclidean gradient $(\sigma \mathbf{I} - \mathbf{A})^{-1}\mathbf{x}_{t-1}$. The key advantage of the preconditioning technique is that we only need to solve the system to an approximate level commensurate with the quality of the current iterate. This can be easily accomplished by performing convex optimization on the associated least-squares problem (see Equation (3)). Another advantage of the reduction to convex optimization is that it enables stochastic optimization [Garber and Hazan, 2015, Garber et al., 2016], especially for the covariance structure of $\mathbf{A} = \frac{1}{m}\mathbf{Y}\mathbf{Y}^\top$, where $\mathbf{Y} \in \mathbb{R}^{m \times d}$. Approximate solutions to the linear systems requires one to cope with inexact Riemannian gradients. In fact, as we will see for Problem (1), the inexact Riemannian gradient method includes the shift-and-inverted power method as a special case with adaptive step-sizes. In the present paper, two other step-size schemes, i.e., constant step-sizes and Barzilai-Borwein (BB) step-sizes, are examined theoretically and/or empirically. Different from Shamir [2015] and Wang et al. [2017] which only consider the positive eigengap between $\lambda_1$ and $\lambda_2$, i.e., $\lambda_1 > \lambda_2$, for the constant step-size setting we explicitly take care of all the cases of this eigengap and achieve a unified convergence rate at $\tilde{O}(\sqrt{\frac{\lambda_1}{\lambda_1 - \lambda_{p+1}}})$ via a novel analysis (e.g., the potential function and the way we cope with the solution space), where $p < n$ is the multiplicity of $\lambda_1$ and $\lambda_1 - \lambda_{p+1} > 0$ always holds without loss of generality. To the best of our knowledge, this is the first time that a gradient descent solver for the problem with fixed step-sizes reaches this type of rate, which is a nearly biquadratic improvement over $\tilde{O}(\frac{1}{(\lambda_1 - \lambda_2)^2})$ [Shamir, 2015]. In addition, the rate logarithmically depends on the initial iterate, instead of quadratically as in Shamir [2015]. Theoretical properties are verified on synthetic data in experiments. For real data, we explore an automatic step-size scheme, i.e, Barzilai-Borwein (BB) step-sizes, to eliminate the difficulty of hand-tuning step-sizes. Experimental results indicate that the shift-and-inverted Riemannian gradient descent method can be significantly faster than the shift-and-inverted power method that has gained much popularity recently.

The rest of the paper is organized as follows. We briefly discuss recent literature in Section 2 and then present our shift-and-inverted Riemannian gradient descent solver with theoretical analysis in Section 3. Experiments are reported in Section 4. The paper then ends with discussions in Section 5.

## 2 Related Work

Recent research on eigenvector computation has been mainly focusing on theoretically scaling up related algorithms. Halko et al. [2011] surveyed and extended randomized algorithms for truncated singular value decomposition (SVD), while Musco and Musco [2015] proposed randomized block Krylov methods for stronger and faster approximate SVD. Convergence rates for both versions are provided in Musco and Musco [2015]. Hardt and Price [2014] studied the noisy power method for the small noise case, and Balcan et al. [2016] extended this method to achieve an improved gap dependency by using subspace iterates of larger dimensions. Garber et al. [2016] presented a robust analysis of the shift-and-invert preconditioned power method and achieved optimal convergence rates. Allen-Zhu and Li [2016] reproved the result for this method and extended to the case that $k > 1$ by deflation via a careful analysis, while Wang et al. [2017] improved the associated analysis and advocated coordinate descent as the solver for linear systems. Lei et al. [2016] proposed a different coordinate-wise power method. Sa et al. [2017] proposed the accelerated (stochastic) power method with optimal rate. However, its empirical performance seems not as good as expected in our experiments. Our work is more related to another line of work on gradient descent solvers. Arora et al. [2012] proposed the stochastic power method without theoretical guarantees which runs the projected stochastic gradient descent (PSGD) for the PCA problem. Arora et al.

Table 1: Typical convergence rates. $\tilde{O}$ notations hide logarithmic factors, e.g., $\log \frac{1}{\epsilon}$, $\log \frac{1}{\lambda_1 - \lambda_2}$.

| Paper | Rate |
|---|---|
| PSGD [Arora et al., 2013] | $O(1/\epsilon^2)$ |
| Oja's algorithm [Balsubramani et al., 2013] | $O(1/((\lambda_1 - \lambda_2)^2 \epsilon))$ |
| Noisy PM [Hardt and Price, 2014] | $\tilde{O}(1/(\lambda_1 - \lambda_2))$ |
| VR-PCA [Shamir, 2015] | $\tilde{O}(1/(\lambda_1 - \lambda_2)^2)$ |
| Power Method (PM) [Musco and Musco, 2015] | $\tilde{O}(1/(\lambda_1 - \lambda_2))$ |
| Block Krylov [Musco and Musco, 2015] | $\tilde{O}(1/\sqrt{\lambda_1 - \lambda_2})$ |
| SGD-PCA [Shamir, 2016b] | $O(1/((\lambda_1 - \lambda_2)\epsilon))$ |
| Shift-and-Inverted PM [Garber et al., 2016] | $\tilde{O}(1/\sqrt{\lambda_1 - \lambda_2})$ |
| Coordiante-wise PM [Lei et al., 2016] | $\tilde{O}(1/(\lambda_1 - \lambda_2))$ |
| Accelerated PM [Sa et al., 2017] | $\tilde{O}(1/\sqrt{\lambda_1 - \lambda_2})$ |
| This work | $\tilde{O}(1/\sqrt{\lambda_1 - \lambda_{p+1}})$ |

[2013] subsequently extended this method via the convex relaxation with theoretical guarantees. Balsubramani et al. [2013] achieved a better guarantee for PCA via the martingale analysis. [Shamir, 2015, 2016a] proposed the VR-PCA which extended the projected stochastic variance reduced gradient (SVRG) to the non-convex PCA problem with global convergence guarantees for the case that $\lambda_1 > \lambda_2$. Shamir [2016b] also studied SGD for the non-convex PCA problem and established its sub-linear convergence rates. Wen and Yin [2013] proposed a practical curvilinear search method for addressing the eigenvalue problem but without theoretical analysis. It actually belongs to the Riemannian gradient descent method. By proving an explicit Łojasiewicz exponent at $\frac{1}{2}$, Liu et al. [2016] established the local and linear convergence rate of the Riemannian gradient method with a line-search procedure for quadratic optimization problems under orthogonality constraints. Details of typical theoretical results are summarized in Table 1.

## 3  Shift-and-Inverted Riemannian Gradient Descent Solver

In this section, we present our shift-and-inverted Riemannian gradient descent solver. Without loss of generality, eigenvalues of the given real symmetric matrix $\mathbf{A}$ are assumed to be in $[0, 1]$ and the multiplicity of the largest eigenvalue $\lambda_1$ is $p$, , i.e., $1 \geq \lambda_1 = \cdots = \lambda_p > \lambda_{p+1} \geq \cdots \geq \lambda_n \geq 0$. Define the $i$-th eigengap of $\mathbf{A}$ as $\Delta_i = \lambda_i - \lambda_{i+1}$. Most of existing work handle only the case that $\Delta_1 = \lambda_1 - \lambda_2 > 0$, ignoring the case that $\Delta_1 = 0$. In this work, the two cases are unified via $\Delta_p > 0$ which holds always without loss of generality[1], i.e., $p < n$. Suppose that corresponding eigenvectors are $\mathbf{v}_1, \cdots, \mathbf{v}_n$. Our goal then is to find one of the leading eigenvectors, i.e., $\mathbf{v} \in \text{span}(\mathbf{v}_1, \cdots, \mathbf{v}_p)$ and $\|\mathbf{v}\|_2 = 1$. Let $\mathbf{V}_p = (\mathbf{v}_1, \cdots, \mathbf{v}_p)$ and denote $\mathbf{B} = (\sigma \mathbf{I} - \mathbf{A})^{-1}$ as the shift-and-inverted matrix, where $\sigma > \lambda_1$. $\mathbf{B}$'s eigenvalues then are $\mu_i = \frac{1}{\sigma - \lambda_i}$ satisfying $\mu_1 = \cdots = \mu_p > \mu_{p+1} \geq \cdots \geq \mu_n$, while eigenvectors remain unchanged. Accordingly, define the $i$-th eigengap of $\mathbf{B}$ as $\tau_i = \mu_i - \mu_{i+1}$. In particular,

$$\frac{\tau_p}{\mu_1} = \frac{\mu_p - \mu_{p+1}}{\mu_1} = \frac{\frac{1}{\sigma - \lambda_p} - \frac{1}{\sigma - \lambda_{p+1}}}{\frac{1}{\sigma - \lambda_1}} = \frac{\Delta_p}{\sigma - \lambda_{p+1}}.$$

A faster rate can be obtained if the relative eigengap can be enlarged from $\mathbf{A}$ to $\mathbf{B}$, which is exactly the idea behind the shift-and-invert preconditioning. To this end, we follow Garber and Hazan [2015] and Wang et al. [2017]'s procedure (see the supplementary material) to choose an appropriate constant $\sigma$ such that it is only slightly larger than $\lambda_1$, i.e., $\sigma = \lambda_1 + c\Delta_p$ where $c \in [\frac{1}{4}, \frac{3}{2}]$, as guaranteed by the following theorem.

**Theorem 3.1** *[Garber and Hazan, 2015, Wang et al., 2017] Let $\epsilon(x) = l(x) - \min_x l(x)$ be the function error with the least-squares subproblem. If the initial to final error ratio for the least-squares subproblems can be maintained as $\frac{\epsilon(\mathbf{z}_0)}{\epsilon(\mathbf{z}^*)} = \frac{32 \cdot 10^{2m} + 1}{\eta^{2m}}$ and $\frac{\epsilon(\mathbf{w}_0)}{\epsilon(\mathbf{w}^*)} = \frac{1024}{\eta^2}$ where $m =$*

$\lceil 8 \log \frac{16}{\|\mathbf{V}_p^\top \tilde{\mathbf{y}}_0\|_2^2} \rceil$, *then we have the output* $\sigma = \lambda_1 + c\Delta_p$ *for certain* $c \in [\frac{1}{4}, \frac{3}{2}]$ *after* $S = O(\log \frac{1}{\eta})$ *iterations in the outer repeat-until loop.*

We then have $\frac{\tau_p}{\mu_1} = \frac{1}{c+1} \geq \frac{2}{5}$ and can run Riemannian gradient descent to solve Problem (1). The algorithmic steps are described in Algorithm 1 which caters to all the step-size settings. Riemannian gradient descent with normalization retraction [Absil et al., 2008], i.e., $R(\mathbf{x}, \xi) = \frac{\mathbf{x}+\xi}{\|\mathbf{x}+\xi\|_2}$ for any $\xi$ from the tangent space of the sphere $\|\mathbf{x}\|_2 = 1$ at $\mathbf{x}$, can be written as

$$
\begin{aligned}
\mathbf{x}_t &= R(\mathbf{x}_{t-1}, -\alpha_t \tilde{\nabla} h(\mathbf{x}_{t-1})) \\
&= R(\mathbf{x}_{t-1}, -\alpha_t(\mathbf{I} - \mathbf{x}_{t-1}\mathbf{x}_{t-1}^\top)\nabla h(\mathbf{x}_{t-1})) \\
&= R(\mathbf{x}_{t-1}, \alpha_t(\mathbf{I} - \mathbf{x}_{t-1}\mathbf{x}_{t-1}^\top)\mathbf{B}\mathbf{x}_{t-1}),
\end{aligned} \tag{2}
$$

where $\tilde{\nabla} h(\mathbf{x}_{t-1})$ and $\nabla h(\mathbf{x}_{t-1})$ represent the Riemannian gradient[2] and Euclidean gradient, respectively. As $(\frac{\mu_1}{\tau_p})^2 = O(1)$, gradient descent takes only a logarithmic number of iterations $O(\log \frac{1}{\epsilon})$ to converge now, which does not have the quadratic dependence on $\frac{\lambda_1}{\lambda_1 - \lambda_2}$ any more [Shamir, 2015, 2016a, Xu et al., 2017, Xu and Gao, 2018, Xu et al., 2018]. However, we need to calculate the Euclidean gradient $\nabla h(\mathbf{x}) = -\mathbf{B}\mathbf{x}$, which by inverting the shifted matrix $\sigma \mathbf{I} - \mathbf{A}$ will be inefficient. Fortunately, as stated in Line 3, Algorithm 1, we can make it efficient by solving an equivalent least-squares subproblem, and an approximate solution to the subproblem will suffice. It is worth noting that when $\alpha_t = 1/\mathbf{x}_t^\top \mathbf{B}\mathbf{x}_t$ Algorithm 1 will recover the shift-and-inverted power method.

---

**Algorithm 1** Shift-and-Inverted Riemannian Gradient Descent Eigensolver

---

1: **Input:** matrix $\mathbf{A}$, shift $\sigma$, and initial $\mathbf{x}_0$.
2: **for** $t = 1, 2, \cdots$ **do**
3:     approximate negative Euclidean gradient

$$
\mathbf{y}_{t-1} \approx \arg\min_{\mathbf{z}} l_t(\mathbf{z}) = \mathbf{z}^\top \mathbf{B}^{-1}\mathbf{z}/2 - \mathbf{x}_{t-1}^\top \mathbf{z} \tag{3}
$$

    by a fast least-squares solver, e.g., AGD, starting from $\mathbf{z}_0 = \mathbf{x}_{t-1}/\mathbf{x}_{t-1}^\top \mathbf{B}^{-1}\mathbf{x}_{t-1}$
4:     approximate Riemannian gradient $\hat{\mathbf{g}}_{t-1} = -(\mathbf{I} - \mathbf{x}_{t-1}\mathbf{x}_{t-1}^\top)\mathbf{y}_{t-1}$
5:     choose a step size $\alpha_t > 0$
6:     set $\mathbf{x}_t = (\mathbf{x}_{t-1} - \alpha_t \hat{\mathbf{g}}_{t-1})/\|\mathbf{x}_{t-1} - \alpha_t \hat{\mathbf{g}}_{t-1}\|_2$
7:     terminate if stopping criterion is met
8: **end for**

---

### 3.1 Analysis

We now provide the convergence analysis of Algorithm 1 under the constant step-size setting. To measure the progress of iterates to one of the leading eigenvectors, we use a novel potential function defined by

$$
\psi(\mathbf{x}_t, \mathbf{V}_p) = -2 \log \|\mathbf{V}_p^\top \mathbf{x}_t\|_2
$$

for analysis. As $\|\mathbf{V}_p^\top \mathbf{x}_t\|_2 \leq \|\mathbf{V}_p\|_2 \|\mathbf{x}_t\|_2 = 1$, we have $\psi(\mathbf{x}_t) \geq 0$. In fact, $\|\mathbf{V}_p^\top \mathbf{x}_t\|_2 = \cos\theta(\mathbf{x}_t, \mathbf{V}_p)$, where $\theta(\mathbf{x}_t, \mathbf{V}_p) \in [0, \frac{\pi}{2}]$ represents the principal angle [Golub and Van Loan, 1996] between $\mathbf{x}_t$ and the space of the leading eigenvectors $\text{span}(\mathbf{V}_p)$. Particularly, it is worth noting that

$$
\theta(\mathbf{x}_t, \mathbf{V}_p) = \min_{\mathbf{v} \in \text{span}(\mathbf{V}_p)} \theta(\mathbf{x}_t, \mathbf{v}),
$$

where $\theta(\mathbf{x}_t, \mathbf{v}) \in [0, \frac{\pi}{2}]$. That is, the angle between a vector $\mathbf{x}$ and a $p$-dimensional subspace $\text{span}(\mathbf{V}_p)$ is equal to the minimum angle between $\mathbf{x}$ and any $\mathbf{v} \in \text{span}(\mathbf{V}_p)$. Thus, we can write

$$
\psi(\mathbf{x}_t, \mathbf{V}_p) = \min_{\mathbf{v} \in \mathcal{V}_{p,1}} \psi(\mathbf{x}_t, \mathbf{v}),
$$

where $\mathcal{V}_{p,1} \triangleq \{\mathbf{v} \in \text{span}(\mathbf{V}_p) : \|\mathbf{v}\|_2 = 1\}$ and $\psi(\mathbf{x}_t, \mathbf{v}) = -2 \log |\mathbf{v}^\top \mathbf{x}_t| = -2 \log \cos\theta(\mathbf{x}_t, \mathbf{v})$ for any $\mathbf{v} \in \mathcal{V}_{p,1}$. This property will play an important role in our analysis. It is easy to see that if

$\psi(\mathbf{x}_t, \mathbf{V}_p)$ goes to 0, $\mathbf{x}_t$ must converge to certain vector $\mathbf{v} \in \mathcal{V}_{p,1}$. We also use another potential function

$$\sin^2 \theta(\mathbf{x}_t, \mathbf{V}_p) = 1 - \|\mathbf{V}_p^\top \mathbf{x}_t\|_2^2.$$

Our main results then can be stated as follows.

**Theorem 3.2** *Given a shift parameter $\sigma = \lambda_1 + c\Delta_1$ for $c \in (0, \frac{3}{2}]$, Algorithm 1 with fixed step-sizes and using accelerated gradient descent as a least-squares solver is able to converge to one of the leading eigenvectors of $\mathbf{A}$, i.e., $\psi(\mathbf{x}_T, \mathbf{V}_p) < \epsilon$, after $T = O(\log \frac{\psi(\mathbf{x}_0, \mathbf{V}_p)}{\epsilon})$ gradient steps, and the overall complexity is $O(\sqrt{\frac{\lambda_1}{\Delta_p}} \log \frac{\lambda_1}{\Delta_p} \log \frac{\psi(\mathbf{x}_0, \mathbf{V}_p)}{\epsilon})$.*

To prove the theorem, we need the following auxiliary lemmas whose proofs are given in the supplementary material.

**Lemma 3.3** $\tau_p \sin^2 \theta(\mathbf{x}, \mathbf{V}_p) \leq \mu_1 - \mathbf{x}^\top \mathbf{B} \mathbf{x} \leq (\mu_1 - \mu_n) \sin^2 \theta(\mathbf{x}, \mathbf{V}_p)$ *and* $\|\tilde{\nabla} h(\mathbf{x})\|_2 \leq 2\mu_1 \sin \theta(\mathbf{x}, \mathbf{V}_p)$.

**Lemma 3.4** $\frac{x}{1+x} \leq \log(1 + x) \leq x$ *for any $x > -1$, while for any $x \in (0,1)$ it holds that* $\frac{x}{-\log(1-x)} \geq \frac{1}{1-\log(1-x)}$.

**Lemma 3.5** *[Wang et al., 2017] Let $\mathbf{z}^\star = \arg\min l_t(\mathbf{z}) = \mathbf{B} \mathbf{x}_{t-1}$, $\xi_t = \mathbf{y}_t - \mathbf{z}^\star$, and $\epsilon_t = l_t(\mathbf{y}_t) - l_t(\mathbf{z}^\star)$. Then $\|\xi_t\|_2 \leq \sqrt{2\mu_1 \epsilon_t}$ and $l_t(\mathbf{z}_0) - l_t(\mathbf{z}^\star) \leq \frac{\mu_1^2}{2\mu_n} \sin^2 \theta(\mathbf{x}_{t-1}, \mathbf{V}_p)$. Moreover, Nesterov's accelerated gradient descent takes $O(\sqrt{\frac{\lambda_1}{\Delta_p}} \log \frac{l_t(\mathbf{z}_0) - l_t(\mathbf{z}^\star)}{\epsilon_t})$ complexity for solving Problem (3) to sub-optimality $\epsilon_t$.*

Since the least-squares solver for Problem (3) is warm-started with $\mathbf{z}_0 = \frac{\mathbf{x}_{t-1}}{\mathbf{x}_{t-1}^\top \mathbf{B}^{-1} \mathbf{x}_{t-1}}$, the initial error $l_t(\mathbf{z}_0) - l_t(\mathbf{z}^\star)$ is much smaller than the error from the random initial $\mathbf{z}_0$. We can also try other least-squares solvers, such as SVRG [Johnson and Zhang, 2013], accelerated SVRG [Garber et al., 2016], and coordinate descent [Wang et al., 2017].

**Proof of Theorem 3.2**

**Proof** For brevity, denote $\theta_t = \theta(\mathbf{x}_t, \mathbf{V}_p)$ and $\psi_t = \psi(\mathbf{x}_t, \mathbf{V}_p)$ throughout the proof. First, for any $\mathbf{v} \in \mathcal{V}_{p,1}$,

$$
\begin{aligned}
\psi(\mathbf{x}_{t+1}, \mathbf{v}) &= -2\log|\mathbf{v}^\top \mathbf{x}_{t+1}| \\
&= -2\log|\mathbf{v}^\top(\mathbf{x}_t - \alpha_{t+1}\hat{\mathbf{g}}_t)| + 2\log\|\mathbf{x}_t - \alpha_{t+1}\hat{\mathbf{g}}_t\|_2.
\end{aligned}
$$

From Lemma 3.5 and Equation (2), we can write $\hat{\mathbf{g}}_t = \tilde{\nabla} h(x_t) - (\mathbf{I} - \mathbf{x}_t \mathbf{x}_t^\top)\xi_t$, where $\xi_t$ is the error with the approximate negative Euclidean gradient in Line 4 of Algorithm 1 incurred from least-squares subproblems (3). We then can expand

$$
\begin{aligned}
&|\mathbf{v}^\top(\mathbf{x}_t - \alpha_{t+1}\hat{\mathbf{g}}_t)|^2 \\
=\ &|\mathbf{v}^\top(\mathbf{x}_t - \alpha_{t+1}\tilde{\nabla}h(x_t)) + \alpha_{t+1}\mathbf{v}^\top(\mathbf{I} - \mathbf{x}_t\mathbf{x}_t^\top)\xi_t|^2 \\
\geq\ &|\mathbf{v}^\top(\mathbf{x}_t - \alpha_{t+1}\tilde{\nabla}h(x_t))|^2 + \alpha_{t+1}^2|\mathbf{v}^\top(\mathbf{I} - \mathbf{x}_t\mathbf{x}_t^\top)\xi_t|^2 \\
&- 2\alpha_{t+1}|\mathbf{v}^\top(\mathbf{x}_t - \alpha_{t+1}\tilde{\nabla}h(x_t))| \cdot |\mathbf{v}^\top(\mathbf{I} - \mathbf{x}_t\mathbf{x}_t^\top)\xi_t| \\
\geq\ &|\mathbf{v}^\top(\mathbf{x}_t - \alpha_{t+1}\tilde{\nabla}h(x_t))|^2 - 2\alpha_{t+1}|\mathbf{v}^\top(\mathbf{x}_t - \alpha_{t+1}\tilde{\nabla}h(x_t))| \cdot |\mathbf{v}^\top(\mathbf{I} - \mathbf{x}_t\mathbf{x}_t^\top)\xi_t| \\
\geq\ &|\mathbf{v}^\top(\mathbf{x}_t - \alpha_{t+1}\tilde{\nabla}h(x_t))|^2 \Big(1 - 2\alpha_{t+1}\frac{\|\mathbf{v}^\top(\mathbf{I} - \mathbf{x}_t\mathbf{x}_t^\top)\|_2\|\xi_t\|_2}{|\mathbf{v}^\top(\mathbf{x}_t - \alpha_{t+1}\tilde{\nabla}h(x_t))|}\Big),
\end{aligned}
\tag{4}
$$

where the last inequality is by the Cauchy-Schwartz inequality. To proceed, we note that

$$\mathbf{v}^\top \tilde{\nabla} h(x_t) = -(\mathbf{v}^\top \mathbf{B}\mathbf{x}_t - \mathbf{v}^\top \mathbf{x}_t \mathbf{x}_t^\top \mathbf{B}\mathbf{x}_t) = -(\mu_1 - \mathbf{x}_t^\top \mathbf{B}\mathbf{x}_t)\mathbf{v}^\top \mathbf{x}_t.$$

Together with Lemma 3.3, we then have

$$
\begin{aligned}
|\mathbf{v}^\top(\mathbf{x}_t - \alpha_{t+1}\tilde{\nabla}h(x_t))| &= (1 + \alpha_{t+1}(\mu_1 - \mathbf{x}_t^\top \mathbf{B}\mathbf{x}_t))|\mathbf{v}^\top \mathbf{x}_t| \\
&\geq (1 + \alpha_{t+1}\tau_p \sin^2 \theta_t)|\mathbf{v}^\top \mathbf{x}_t|.
\end{aligned}
\tag{5}
$$

In addition, one can write

$$\|\mathbf{v}^\top(\mathbf{I} - \mathbf{x}_t\mathbf{x}_t^\top)\|_2 = \|\mathbf{v}^\top\mathbf{x}_t^\perp\|_2 = (\mathbf{v}^\top\mathbf{x}_t^\perp(\mathbf{x}_t^\perp)^\top\mathbf{v})^{1/2} = (\mathbf{v}^\top(\mathbf{I} - \mathbf{x}_t\mathbf{x}_t^\top)\mathbf{v})^{1/2}$$
$$= (1 - (\mathbf{v}^\top\mathbf{x}_t)^2)^{1/2} = \sin\theta(\mathbf{x}_t, \mathbf{v}), \tag{6}$$

and

$$\|\mathbf{x}_t - \alpha_{t+1}\hat{g}_t\|_2^2 = (\mathbf{x}_t - \alpha_{t+1}\hat{\mathbf{g}}_t)^\top(\mathbf{x}_t - \alpha_{t+1}\hat{\mathbf{g}}_t) = 1 + \alpha_{t+1}^2\|\hat{\mathbf{g}}_t\|_2^2$$
$$\leq 1 + 2\alpha_{t+1}^2(\|\tilde{\nabla}h(x_t)\|_2^2 + \|\xi_t\|_2^2) \leq 1 + 2\alpha_{t+1}^2(4\mu_1^2\sin^2\theta_t + \|\xi_t\|_2^2), \tag{7}$$

where the last inequality is due to Lemma 3.3. By (4)-(7), one can arrive at

$$\psi(\mathbf{x}_{t+1}, \mathbf{v}) \leq \psi(\mathbf{x}_t, \mathbf{v}) - 2\log(1 + \alpha_{t+1}\tau_p\sin^2\theta_t) - \log(1 - \frac{2\alpha_{t+1}\|\xi_t\|_2\tan\theta(\mathbf{x}_t, \mathbf{v})}{1 + \alpha_{t+1}\tau_p\sin^2\theta_t})$$
$$+ \log(1 + 2\alpha_{t+1}^2(4\mu_1^2\sin^2\theta_t + \|\xi_t\|_2^2)).$$

Taking the minimum with respect to $\mathbf{v}$ over $\mathcal{V}_{p,1}$ on both sides and noting that $\|\xi_t\|_2 \leq \sqrt{2\mu_1\epsilon_t}$ by Lemma 3.5, we then get

$$\psi_{t+1} \leq \psi_t - 2\log(1 + \alpha_{t+1}\tau_p\sin^2\theta_t) - \log(1 - \frac{2\alpha_{t+1}\sqrt{2\mu_1\epsilon_t}\tan\theta_t}{1 + \alpha_{t+1}\tau_p\sin^2\theta_t})$$
$$+ \log(1 + 2\alpha_{t+1}^2(4\mu_1^2\sin^2\theta_t + 2\mu_1\epsilon_t)).$$

Letting $\epsilon_t = \frac{\tau_p^2}{\mu_1}\frac{\sin^2(2\theta_t)}{32}$, the above inequality can be reduced:

$$\psi_{t+1} \leq \psi_t - 2\log(1 + \alpha_{t+1}\tau_p\sin^2\theta_t) - \log(1 - \frac{\alpha_{t+1}\tau_p\sin^2\theta_t}{1 + \alpha_{t+1}\tau_p\sin^2\theta_t})$$
$$+ \log(1 + 2\alpha_{t+1}^2(4\mu_1^2\sin^2\theta_t + \frac{\tau_p^2}{4}\sin^2\theta_t\cos^2\theta_t))$$
$$\leq \psi_t - \log(1 + \alpha_{t+1}\tau_p\sin^2\theta_t) + \log(1 + 10\alpha_{t+1}^2\mu_1^2\sin^2\theta_t)$$
$$\leq \psi_t - \frac{\alpha_{t+1}\tau_p\sin^2\theta_t}{1 + \alpha_{t+1}\tau_p\sin^2\theta_t} + 10\alpha_{t+1}^2\mu_1^2\sin^2\theta_t) \quad \text{(by Lemma 3.4)}$$
$$\leq \psi_t - \alpha_{t+1}(\frac{\tau_p}{1 + \alpha_{t+1}\tau_p} - 10\alpha_{t+1}\mu_1^2)\sin^2\theta_t.$$

Thus, if $\frac{\tau_p}{2(1+\alpha_{t+1}\tau_p)} - 10\alpha_{t+1}\mu_1^2 > 0$, i.e., $\alpha_{t+1} < \frac{\tau_p}{20\mu_1^2(1+\alpha_{t+1}\tau_p)}$, we then get $\psi_{t+1} < \psi_t$ and $\psi_{t+1} \leq \psi_t - \frac{\alpha_{t+1}\tau_p\sin^2\theta_t}{2(1+\alpha_{t+1}\tau_p)}$. Note that

$$\sin^2\theta_t = \frac{\sin^2\theta_t}{-\log(1 - \sin^2\theta_t)}\cdot\psi_t \geq \frac{\psi_t}{1 - \log(1 - \sin^2\theta_t)} = \frac{\psi_t}{1 + \psi_t} \geq \frac{\psi_t}{1 + \psi_0},$$

where the first inequality is by Lemma 3.4. If $\alpha_t \equiv \alpha$, we then can arrive at

$$\psi_T \leq (1 - \frac{\alpha\tau_p}{2(1 + \alpha\tau_p)}\cdot\frac{1}{1 + \psi_0})\psi_{T-1} \leq (1 - \frac{\alpha\tau_p}{2(1 + \alpha\tau_p)}\cdot\frac{1}{1 + \psi_0})^T\psi_0$$
$$\leq \exp\{-T\frac{\alpha\tau_p}{2(1 + \alpha\tau_p)}\cdot\frac{1}{1 + \psi_0}\}\psi_0 \triangleq \Xi.$$

Setting $\Xi = \epsilon$ and noting $\alpha < \frac{\tau_p}{20\mu_1^2(1+\alpha\tau_p)}$ yields

$$T = \frac{2(1 + \alpha\tau_p)(1 + \psi_0)}{\alpha\tau_p}\log\frac{\psi_0}{\epsilon} = O(\frac{1}{\alpha\tau_p}\log\frac{\psi_0}{\epsilon}) = O((\frac{\mu_1}{\tau_p})^2\log\frac{\psi_0}{\epsilon}) = O(\log\frac{\psi_0}{\epsilon}).$$

On the other hand, by Lemma 3.5, the complexity for computing $\mathbf{y}_t$ is

$$O(\sqrt{\frac{\lambda_1}{\Delta_p}}\log\frac{l_t(\mathbf{z}_0) - l_t(\mathbf{Bx}_{t-1})}{\epsilon_t}) = O(\sqrt{\frac{\lambda_1}{\Delta_p}}\log\frac{\frac{\mu_1^2}{\mu_n}\sin^2\theta_t}{\frac{\tau_p^2}{\mu_1}\frac{\sin^2(2\theta_t)}{32}})$$
$$= O(\sqrt{\frac{\lambda_1}{\Delta_p}}(\log\frac{\mu_1}{\mu_n} + \psi_t)) = O(\sqrt{\frac{\lambda_1}{\Delta_p}}(\log\frac{\mu_1}{\mu_n} + \psi_0)) = O(\sqrt{\frac{\lambda_1}{\Delta_p}}\log\frac{\lambda_1}{\Delta_p}),$$

Thus, the overall complexity is $O(\sqrt{\frac{\lambda_1}{\Delta_p}}\log\frac{\lambda_1}{\Delta_p}\log\frac{\psi_0}{\epsilon}) = \tilde{O}(\sqrt{\frac{\lambda_1}{\Delta_p}})$. $\qquad\square$

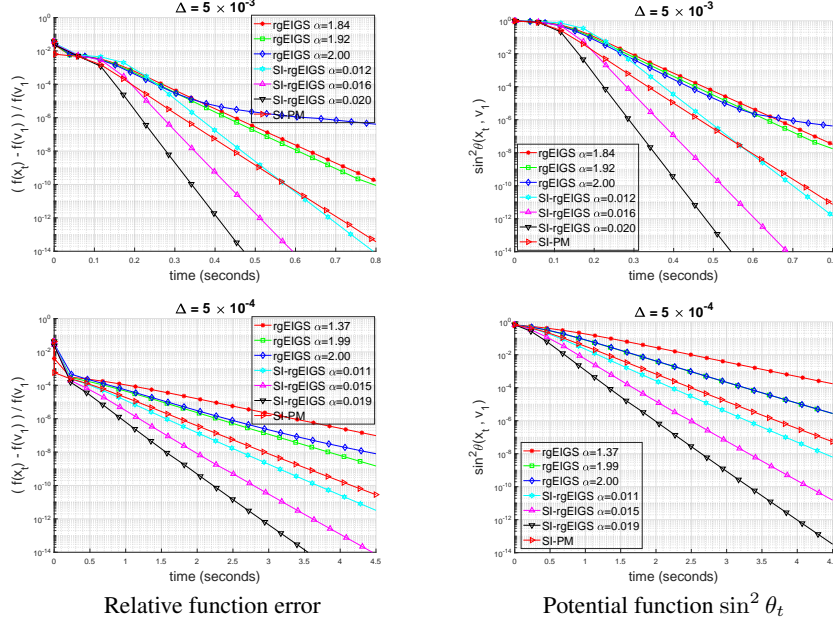

Relative function error             Potential function $\sin^2 \theta_t$

Figure 1: Synthetic data.

# 4 Experiments

We test our algorithm on both synthetic and real data. Throughout experiments, our SI-rgEIGS solver is warm-started by a few power iterations, and four iterations of Nesterov's AGD are run to approximately solve the least-squares subproblems. The same initial $\mathbf{x}_0$ is used for different solvers. All the algorithms are implemented in matlab and running single threaded. All the ground-truth information is obtained by matlab's eigs function for benchmarking purpose. The implementation of our algorithm is available at `https://github.com/zhiqiangxu2001/SI-rgEIGS`.

## 4.1 Synthetic Data

We follow Shamir [2015] to generate synthetic data. Note that $\mathbf{A}$'s full eigenvalue decomposition can be written as $\mathbf{A} = \mathbf{V}_n \boldsymbol{\Sigma} \mathbf{V}_n^\top$, where $\boldsymbol{\Sigma}$ is diagonal. Thus, it suffices to generate random orthogonal matrix $\mathbf{V}_n$ and set $\boldsymbol{\Sigma} = \mathrm{diag}(1, 1 - \Delta, 1 - 1.1\Delta, \cdots, 1 - 1.4\Delta, g_1/n, \cdots, g_{n-6}/n)$ with $g_i$ being standard normal samples, i.e., $g_i \sim \mathcal{N}(0, 1)$. Here we set $n = 1000$ and $\sigma = 1.005$, and three solvers are compared: Rimennian gradient descent solver with/without shift-and-invert preconditioning under the constant step-size setting, and the shift-and-inverted power method [Garber et al., 2016]. Constant step-sizes are hand-tuned. Figure 1 reports the performance of three algorithms, in terms of relative function error $(f(\mathbf{x}_t) - f(\mathbf{v}_1))/f(\mathbf{v}_1)$ or the potential $\sin^2 \theta_t$, where we use $f(\mathbf{x}) = \mathbf{x}^\top \mathbf{A} \mathbf{x}$ and then $f(\mathbf{v}_1) = \lambda_1 = \max_{\mathbf{x} \in \mathbb{R}^{n \times 1}: \|\mathbf{x}\|_2 = 1} f(\mathbf{x})$. We see that Riemannian gradient descent with shift-and-invert preconditioning indeed outperforms the counterpart without preconditioning which is also worse than the SI-PM. This demonstrates the effectiveness of the shift-and-invert preconditioning for acceleration again. Second, unexpectedly, SI-rgEIGS runs faster than SI-PM, despite an extra log factor in theory. This may hint at the possibility of removing this factor in analysis of our method. Last, note that convergence behaviors are consistent in terms of two quality measures.

## 4.2 Real Data

We now demonstrate the performance of Algorithm 1 on real data from the sparse matrix collection, and also compare with the accelerated power method with optimal momentum $\beta = \lambda_2^2/4$ (abbreviated as APM-OM) [Sa et al., 2017]. However, two issues need to be fixed. First, the crude phase of Garber and Hazan [2015], Wang et al. [2017] for locating the shift parameter is hard to use as there

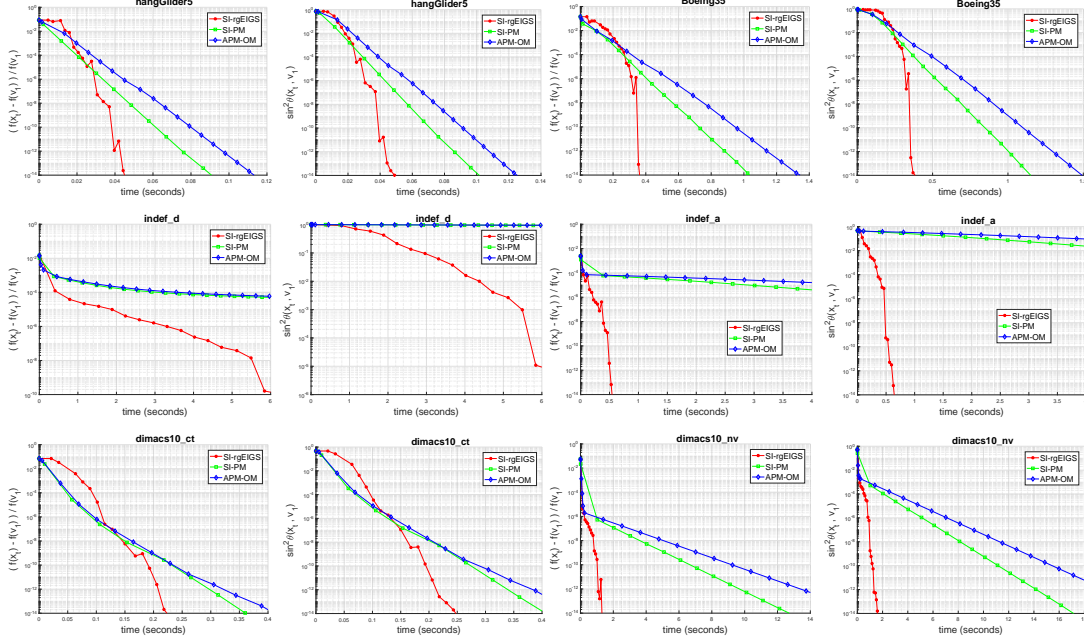

Figure 2: Real data.

are three parameters that need to be tuned. We use heuristics based on Lemma 3.3. The lemma shows that $\lambda_1 \leq \mathbf{x}_t^\top \mathbf{A} \mathbf{x}_t + (\lambda_1 - \lambda_n) \sin^2 \theta_t$ and $\|\tilde{\nabla} f(\mathbf{x}_t)\|_2 \leq 2\lambda_1 \sin \theta_t$. Then we have the upper bound on $\lambda_1$: $\sigma = \mathbf{x}_{T_c}^\top \mathbf{A} \mathbf{x}_{T_c} + \beta \|\tilde{\nabla} f(\mathbf{x}_{T_c})\|_2^2$ for proper constant $\beta > \frac{1}{2}$ and wart-start $\mathbf{x}_{T_c}$ from the crude phrase. We find that setting $\beta = 1/\|\tilde{\nabla} f(\mathbf{x}_{T_c})\|_2$ works well on our data. Second, hand-tuning of step-sizes, even for constant step-sizes, is a difficult task. We thus use an automatic step-size scheme, specifically, Barzilai-Borwein (BB) step-size, which is a non-monotone step-size scheme and performs well in practice [Wen and Yin, 2013]. In our context, it is set as follows:

$$\alpha_{t+1} = \frac{\|\mathbf{x}_t - \mathbf{x}_{t-1}\|_2^2}{|(\mathbf{x}_t - \mathbf{x}_{t-1})^\top (\hat{g}_t - \hat{g}_{t-1})|}, \quad \text{or} \quad \alpha_{t+1} = \frac{|(\mathbf{x}_t - \mathbf{x}_{t-1})^\top (\hat{g}_t - \hat{g}_{t-1})|}{\|\hat{g}_t - \hat{g}_{t-1}\|_2^2}.$$

Note that we use inexact Riemannian gradients $\hat{g}_t$ here, instead of exact ones $\tilde{\nabla} h(\mathbf{x})$ as in the traditional case. Nonetheless, it still performs well and significantly better than the shift-and-inverted power method as observed in Figure 2. See the supplementary material for the description of the real data.

## 5 Discussions

In this work, we investigated Riemannian gradient descent with shift-and-invert preconditioning for the leading eigenvector computation on the effect of step-size schemes, in comparison to the recently popular shift-and-inverted power method. Specifically, the constant step-size scheme and the Barzilai-Borwein (BB) step-size scheme were considered theoretically and/or empirically. The algorithm was theoretically analyzed under the constant step-size setting and shown for the first time to able to achieve a rate of the type $\tilde{O}\left(\sqrt{\frac{\lambda_1}{\lambda_1 - \lambda_{p+1}}}\right)$ and a logarithmic dependence on the initial iterate. It is a nearly biquadratic improvement for the gradient descent solver, covering both $\Delta_1 > 0$ and $\Delta_1 = 0$. Experimental results demonstrated that the shift-and-invert preconditioning can indeed accelerate gradient descent solver. Unexpectedly, the adaptive step-size setting with the shift-and-inverted power method is outperformed by the considered step-size settings, especially the BB step-size scheme on real data, albeit with a provable optimal rate. For future work, we may further investigate if the log factor $\log \frac{\lambda_1}{\lambda_1 - \lambda_{p+1}}$ can be removed from the overall complexity and test our algorithms with other least-squares solvers for deeper understanding of its performance.

## Acknowledgments

Authors would like to thank the reviewers, AC, and SAC for their valuable comments.

## Footnotes

[1]If $p = n$, the objective function $h(\mathbf{x})$ is constant and Problem (1) is trivial.

[2] It can be obtained by projecting the Euclidean gradient onto the tangent space [Absil et al., 2008] at $\mathbf{x}_{t-1}$, i.e., $\tilde{\nabla} h(\mathbf{x}_{t-1}) = \mathbf{P}_{\mathbf{x}_{t-1}} \nabla h(\mathbf{x}_{t-1})$, where $\mathbf{P}_{\mathbf{x}_{t-1}} = \mathbf{I} - \mathbf{x}_{t-1}\mathbf{x}_{t-1}^\top$.

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
