[Supplementary Material · nips2018xu _supp.pdf]

# Supplementary Material: Gradient Descent Meets Shift-and-Invert Preconditioning for Eigenvector Computation

**Zhiqiang Xu**
Cognitive Computing Lab (CCL), Baidu Research
National Engineering Laboratory of Deep Learning Technology and Application, China
xuzhiqiang04@baidu.com

**Shift Parameter**   Garber and Hazan [2015] and Wang et al. [2017]'s procedure to locate a shift parameter $\sigma$ is described in Algorithm 1.

**Lemma 1**   $\tau_p \sin^2 \theta(\mathbf{x}, \mathbf{V}_p) \leq \mu_1 - \mathbf{x}^\top \mathbf{B} \mathbf{x} \leq (\mu_1 - \mu_n) \sin^2 \theta(\mathbf{x}, \mathbf{V}_p)$ and $\|\tilde{\nabla} h(\mathbf{x})\|_2 \leq 2\mu_1 \sin \theta(\mathbf{x}, \mathbf{V}_p)$.

**Proof**   I) Write the full eigenvalue decomposition of $\mathbf{B}$ as

$$\mathbf{B} = \mu_1 \mathbf{V}_p \mathbf{V}_p^\top + \mathbf{V}_p^\perp \mathrm{diag}(\mu_{p+1}, \cdots, \mu_n)(\mathbf{V}_p^\perp)^\top, \tag{1}$$

where $\mathbf{V}_p^\perp$ represents the orthogonal complement of $\mathbf{V}_p$. One then has

$$
\begin{aligned}
\mu_1 - \mathbf{x}^\top \mathbf{B} \mathbf{x} &= \mu_1 - \mu_1 \mathbf{x}^\top \mathbf{V}_p \mathbf{V}_p^\top \mathbf{x} - \mathbf{x}^\top \mathbf{V}_p^\perp \mathrm{diag}(\mu_{p+1}, \cdots, \mu_n)(\mathbf{V}_p^\perp)^\top \mathbf{x} \\
&\geq \mu_1 \sin^2 \theta(\mathbf{x}, \mathbf{V}_p) - \mu_{p+1} \mathbf{x}^\top \mathbf{V}_p^\perp (\mathbf{V}_p^\perp)^\top \mathbf{x} \\
&= \mu_1 \sin^2 \theta(\mathbf{x}, \mathbf{V}_p) - \mu_{p+1} \mathbf{x}^\top (\mathbf{I} - \mathbf{V}_p \mathbf{V}_p^\top) \mathbf{x} \\
&= \tau_p \sin^2 \theta(\mathbf{x}, \mathbf{V}_p).
\end{aligned}
$$

On the other hand,

$$
\begin{aligned}
\mu_1 - \mathbf{x}^\top \mathbf{B} \mathbf{x} &= \mu_1 - \mu_1 \mathbf{x}^\top \mathbf{V}_p \mathbf{V}_p^\top \mathbf{x} - \mathbf{x}^\top \mathbf{V}_p^\perp \mathrm{diag}(\mu_{p+1}, \cdots, \mu_n)(\mathbf{V}_p^\perp)^\top \mathbf{x} \\
&\leq \mu_1 \sin^2 \theta(\mathbf{x}, \mathbf{V}_p) - \mu_n \mathbf{x}^\top \mathbf{V}_p^\perp (\mathbf{V}_p^\perp)^\top \mathbf{x} \\
&= \mu_1 \sin^2 \theta(\mathbf{x}, \mathbf{V}_p) - \mu_n \mathbf{x}^\top (\mathbf{I} - \mathbf{V}_p \mathbf{V}_p^\top) \mathbf{x} \\
&= (\mu_1 - \mu_n) \sin^2 \theta(\mathbf{x}, \mathbf{V}_p).
\end{aligned}
$$

II) For any $\mathbf{v} \in \mathcal{V}_{p,1}$, we can write $\mathbf{B} = \mu_1 \mathbf{v} \mathbf{v}^\top + \mathbf{v}_\perp \mathrm{diag}(\mu_2, \cdots, \mu_n) \mathbf{v}_\perp^\top$. Plugging in the above equation to the gradient, one gets

$$
\begin{aligned}
\|\tilde{\nabla} h(\mathbf{x})\|_2^2 &= \|(\mathbf{I} - \mathbf{x} \mathbf{x}^\top) \mathbf{B} \mathbf{x}\|_2^2 = \|\mathbf{x}_\perp^\top (\mu_1 \mathbf{v} \mathbf{v}^\top + \mathbf{v}_\perp \mathrm{diag}(\mu_2, \cdots, \mu_n) \mathbf{v}_\perp^\top) \mathbf{x}\|_2^2 \\
&\leq 2\mu_1^2 \|\mathbf{x}_\perp^\top \mathbf{v}\|^2 + 2\mu_2^2 \|\mathbf{v}_\perp^\top \mathbf{x}\|^2 = 2\mu_1^2 (1 - (\mathbf{x}^\top \mathbf{v})^2) + 2\mu_2^2 (1 - (\mathbf{v}^\top \mathbf{x})^2) \\
&\leq 4\mu_1^2 (1 - (\mathbf{x}^\top \mathbf{v})^2).
\end{aligned}
$$

Since the above inequality holds for any $\mathbf{v} \in \mathcal{V}_{p,1}$, we get

$$\|\tilde{\nabla} h(\mathbf{x})\|_2^2 \leq 4\mu_1^2 \min_{\mathbf{v} \in \mathcal{V}_{p,1}} (1 - (\mathbf{x}^\top \mathbf{v})^2) = 4\mu_1^2 \sin^2 \theta(\mathbf{x}, \mathbf{V}_p).$$

$\square$

**Lemma 2**   $\frac{x}{1+x} \leq \log(1+x) \leq x$ for any $x > -1$, while for any $x \in (0,1)$ it holds that $\frac{x}{-\log(1-x)} \geq \frac{1}{1-\log(1-x)}$.

**Algorithm 1** [Garber and Hazan, 2015, Wang et al., 2017] locate $\sigma = \lambda_1 + c\Delta_p$

---

1: **Input:** matrix $\mathbf{A}$ and lower estimate $\eta$ satisfying $c_1\Delta_p \leq \eta \leq c_2\Delta_p$ where $0 < c_1 < c_2 \leq 1$.
2: $\tilde{\mathbf{y}}_0 = \mathbf{u}/\|\mathbf{u}\|_2$ where $\mathbf{u} \in \mathbb{R}^{n \times 1}$ and $\mathbf{u}_i \sim \mathcal{N}(0, 1)$
3: $s = 0$ and $\sigma_s = 1 + \eta$
4: **repeat**
5:     $\mathbf{y}_0 = \tilde{\mathbf{y}}_s$
6:     **for** $t = 1, 2, \cdots, m$ **do**
7:         $\mathbf{z}^* \approx \arg\min_{\mathbf{z}} \frac{1}{2}\mathbf{z}^\top(\sigma_s\mathbf{I} - \mathbf{A})\mathbf{z} - \mathbf{y}_{t-1}^\top\mathbf{z}$ starting from $\mathbf{z}_0 = \frac{\mathbf{y}_{t-1}}{\mathbf{y}_{t-1}^\top(\sigma_s\mathbf{I}-\mathbf{A})\mathbf{y}_{t-1}}$
8:         $\mathbf{y}_t = \mathbf{z}^*/\|\mathbf{z}^*\|_2$
9:     **end for**
10:    $\tilde{\mathbf{y}}_{s+1} = \mathbf{y}_m$
11:    $\mathbf{w}^* \approx \arg\min_{\mathbf{w}} \frac{1}{2}\mathbf{w}^\top(\sigma_s\mathbf{I} - \mathbf{A})\mathbf{w} - \tilde{\mathbf{y}}_{s+1}^\top\mathbf{w}$ starting from $\mathbf{w}_0 = \frac{\tilde{\mathbf{y}}_{s+1}}{\tilde{\mathbf{y}}_{s+1}^\top(\sigma_s\mathbf{I}-\mathbf{A})\tilde{\mathbf{y}}_{s+1}}$
12:    $\eta_{s+1} = \frac{1}{2}\frac{1}{\tilde{\mathbf{y}}_{s+1}^\top\mathbf{w}^\star - \frac{1}{8}(1 + \frac{1-c_2}{c_2}\eta)}$ and $\sigma_{s+1} = \sigma_s - \frac{1}{2}\eta_{s+1}$
13:    $s \leftarrow s + 1$
14: **until** $\eta_s \leq \eta$
15: **Output:** $\sigma = \sigma_s$ and $\mathbf{x}_0 = \tilde{\mathbf{y}}_s$

---

**Proof**   1) For any $x$, it holds that $1 + x \leq e^x$. Then for any $x > -1$,

$$\log(1 + x) \leq x.$$

If one lets $y = 1 + x$ in the above inequality, then $\log y \leq y - 1$. Further letting $y = \frac{1}{z}$ yields $\log z \geq -\frac{1}{z} + 1 = \frac{z-1}{z}$. Last, setting $z = 1 + x$ gives us

$$\log(1 + x) \geq \frac{x}{1 + x}.$$

2) Note that $\log(1 + x) = \sum_{i=0}^\infty (-1)^i \frac{x^{i+1}}{i+1}$ for $|x| < 1$. One then can write for $x \in (0, 1)$ that

$$
\begin{aligned}
\frac{x}{-\log(1 - x)} &= \frac{x}{-\sum_{i=0}^\infty(-1)^i\frac{(-x)^{i+1}}{i+1}} = \frac{1}{\sum_{i=0}^\infty\frac{x^i}{i+1}} = \frac{1}{1 + \sum_{i=1}^\infty\frac{x^i}{i+1}} \\
&\geq \frac{1}{1 + \sum_{i=1}^\infty\frac{x^i}{i}} = \frac{1}{1 - \sum_{i=1}^\infty(-1)^{i-1}\frac{(-x)^i}{i}} = \frac{1}{1 - \log(1 - x)}.
\end{aligned}
$$

$\qquad\qquad\qquad\qquad\qquad\qquad\qquad\qquad\qquad\qquad\qquad\qquad\qquad\qquad\qquad\qquad\qquad\qquad\square$

**Lemma 3**   [Wang et al., 2017] Let $\mathbf{z}^\star = \arg\min l_t(\mathbf{z}) = \mathbf{B}\mathbf{x}_{t-1}$, $\xi_t = \mathbf{y}_t - \mathbf{z}^\star$, and $\epsilon_t = l_t(\mathbf{y}_t) - l_t(\mathbf{z}^\star)$. Then $\|\xi_t\|_2 \leq \sqrt{2\mu_1\epsilon_t}$ and $l_t(\mathbf{z}_0) - l_t(\mathbf{z}^\star) \leq \frac{\mu_1^2}{2\mu_n}\sin^2\theta(\mathbf{x}_{t-1}, \mathbf{V}_p)$. Moreover, Nesterov's accelerated gradient descent takes $O(\sqrt{\frac{\lambda_1}{\Delta_p}}\log\frac{l_t(\mathbf{z}_0)-l_t(\mathbf{z}^\star)}{\epsilon_t})$ complexity for solving Problem (4) (in the main text) to sub-optimality $\epsilon_t$.

**Proof**   I) The proof can be found in [Wang et al., 2017] and is included here with slight modification. For the quadratic function $l_t(\mathbf{z})$, we can write

$$\epsilon(\mathbf{z}) = l_t(\mathbf{z}) - l_t(\mathbf{z}^\star) = \frac{1}{2}(\mathbf{z} - \mathbf{z}^\star)^\top\mathbf{B}^{-1}(\mathbf{z} - \mathbf{z}^\star) = \frac{1}{2}\|\mathbf{z} - \mathbf{z}^\star\|_{\mathbf{B}^{-1}}^2.$$

Thus,

$$
\begin{aligned}
\|\xi_t\|_2 &= \|\mathbf{y}_t - \mathbf{z}^\star\|_2 = \|\mathbf{B}^{\frac{1}{2}}\mathbf{B}^{-\frac{1}{2}}(\mathbf{y}_t - \mathbf{z}^\star)\|_2 \\
&\leq \|\mathbf{B}^{\frac{1}{2}}\|_2\|\mathbf{B}^{-\frac{1}{2}}(\mathbf{y}_t - \mathbf{z}^\star)\|_2 = \sqrt{\mu_1}\|\mathbf{y}_t - \mathbf{z}^\star\|_{\mathbf{B}^{-1}} \\
&= \sqrt{2\mu_1\epsilon_t}.
\end{aligned}
$$

Note that

$$s(\gamma) = \epsilon(\gamma\mathbf{x}_{t-1}) = l_t(\gamma\mathbf{x}_{t-1}) - l_t(\mathbf{z}^\star) = \frac{\gamma^2}{2}\mathbf{x}_{t-1}^\top\mathbf{B}^{-1}\mathbf{x}_{t-1} - \gamma - l_t(\mathbf{z}^\star),$$

Figure 1: Synthetic data.

Table 1: Statistics of the data.

| Matrix | n | # nonzero entries |
|---|---|---|
| hangGlider5 | 16011 | 155246 |
| Boeing35 | 30237 | 1450163 |
| indef_d | 60000 | 299998 |
| indef_a | 60008 | 255004 |
| dimacs10_ct | 67578 | 336352 |
| dimacs10_nv | 84538 | 416998 |

which is minimized at $\gamma = \frac{1}{\mathbf{x}_{t-1}^\top \mathbf{B}^{-1}\mathbf{x}_{t-1}}$. Thus, we have

$$
\begin{aligned}
l_t(\mathbf{z}_0) - l_t(\mathbf{z}^\star) &\leq & l_t(\mu_1 \mathbf{x}_{t-1}) - l_t(\mathbf{z}^\star) = \frac{\mu_1^2}{2}\mathbf{x}_{t-1}^\top \mathbf{B}^{-1}\mathbf{x}_{t-1} - \mu_1 - l_t(\mathbf{z}^\star) \\
&=& \frac{\mu_1^2}{2}\sum_{i=1}^n \frac{(\mathbf{v}_i^\top \mathbf{x}_{t-1})^2}{\mu_i} - \mu_1 \sum_{i=1}^n (\mathbf{v}_i^\top \mathbf{x}_{t-1})^2 + \frac{1}{2}\sum_{i=1}^n \mu_i (\mathbf{v}_i^\top \mathbf{x}_{t-1})^2 \\
&\leq& \frac{1}{2}\sum_{i=1}^n \frac{(\mu_1 - \mu_i)^2}{\mu_i}(\mathbf{v}_i^\top \mathbf{x}_{t-1})^2 \leq \frac{\mu_1^2}{2\mu_n}\sum_{i=p+1}^n (\mathbf{v}_i^\top \mathbf{x}_{t-1})^2 \\
&=& \frac{\mu_1^2}{2\mu_n}(1 - \sum_{i=1}^p (\mathbf{v}_i^\top \mathbf{x}_{t-1})^2) = \frac{\mu_1^2}{2\mu_n}(1 - \|\mathbf{V}_p^\top \mathbf{x}_{t-1}\|^2) \\
&=& \frac{\mu_1^2}{2\mu_n}\sin^2\theta(\mathbf{x}_{t-1}, \mathbf{V}_p).
\end{aligned}
$$

II) The complexity can be obtained by noting that the Hessian of $l_t(\mathbf{z})$ satisfies

$$
\frac{1}{\mu_1}\mathbf{I} \preccurlyeq \text{Hessian}(l_t(\mathbf{z})) = \mathbf{B}^{-1} \preccurlyeq \frac{1}{\mu_n}\mathbf{I}.
$$

That is, $l_t(\mathbf{z})$ is $\frac{1}{\mu_1}$-strongly convex and $\frac{1}{\mu_n}$-smooth. Thus, Nesterov's accelerated gradient descent takes

$$
O\left(\sqrt{\frac{\frac{1}{\mu_n}}{\frac{1}{\mu_1}}}\log\frac{l_t(\mathbf{z}_0) - l_t(\mathbf{z}^\star)}{\epsilon_t}\right) = O\left(\sqrt{\frac{\lambda_1}{\Delta_p}}\log\frac{l_t(\mathbf{z}_0) - l_t(\mathbf{z}^\star)}{\epsilon_t}\right)
$$

complexity to reach suboptimality $\epsilon_t$. □

**Robustness of $\sigma$** We test the robustness of $\sigma$ on the synthetic data with $\Delta = 5\times10^{-3}$. Performance of the algorithm with varying $\sigma$ and best-tuned constant step-size is shown in Figure 1. We see that smaller $\sigma$ yields faster convergence.

**Real Data** It can be found at `www.cise.ufl.edu/research/sparse/matrices/`. See Table 1 for the statistics of the data.