[Reviews · NeurIPS 2018]

Reviewer 1



The main idea is incorporating Nesterov's accelerated gradient descent (AGD) in eigenvalue problem. The approach relies on shift-and-invert preconditioning method that reduces the non-convex objective of Rayleigh quotient to a sequence of convex programs. Shift-and-invert preconditioning improves the convergence dependency of the gradient method to the eigengap of the given matrix. The focus of this paper is using AGD method to approximately solve the convex programs and reaching an accelerated convergence rate for the convex part. Exploiting the accelerated convergence of AGD, they reach an accelerated convergence for the first-order optimization of the eigenvalue problem. My main concern is regarding the novelty of the paper. There is an extensive literature on accelerating power iteration method that is totally neglected in this paper. For example, "accelerated stochastic power iteration" by CHRISTOPHER DE SA et al. is a recent one. Chebyshev's acceleration in the classic literature of optimization of Rayleigh quotient is also quite related to this paper (see "convergence rate estimates for iterative methods for a mesh symmetric eigenvalue problem" by Knyazev). Indeed, gradient descent on eigenvalue problem can directly be accelerated without the "shift-and-invert" convex relaxation. The accelerated power iteration does not have the extra logarithmic dependency to the eigengap. In section 3, the authors are motivating the advantage of their approach in the case of the multiplicity of the largest eigenvalues. However, I think that this case is not considered in the literature due to the degeneracy. As it is mentioned in the introduction, eigenvector problem search for the leading eigenvector. Since the leading eigenvector is not unique in the multiplicity case, this case is skipped in literature. In such a degenerate case, one can still provide a convergence on the eigenvalue (function f(x)) but the leading eigenvector is not well-defined. The required conditions on initial parameter x_0 are missed in the theorem as well as the algorithm. It is clear that if x_0 is orthogonal to the subspace V, then the algorithm can not retrieve the desired global minimum. To make the algorithm clearer, I suggest making the terms such that approximate gradient more precise. I think that the number of AGD iterations in step 3 can be determined by their analysis, so it might be better to include this number in the Algorithm as well. Since the shift parameter sigma is unknown, I think they should include the time complexity of searching for this parameter and analyze the robustness of the method against the choice of sigma. Regarding experiments, I think that one baseline on real datasets is not enough. Particularly, eigenvector problem is one of the oldest optimization problems with many solvers. For example, accelerated power methods can be a potential baseline for their proposal. ------------------------------------------------------- I have read the response. Authors had convincing responses and experiements regarding the literature and baselines, hence I decided to increase my score by one. Still, I think that handling the degenerated case lambda_1 = lambda_2 is not a major contribution.

Reviewer 2



The paper proposes a complex combination of: -- shift-and-inverted power method [Garber and Hazan, 2015] -- shift-and-invert pre-conditioning using thus obtained shift parameter -- gradient descent for leading eigenvector computations. The authors try to prove the method can match the convergence rate of Lanczos. This relies on a number of parameters incl. a lower estimate of the eigengap and two iteration numbers (cf. line 60), and a stopping criterion for the first step, presumably Lipschitz constant estimates for the third (or a method for estimating it) or some additional parameters of the accelerated algorithm, etc. This improves considerably upon [1] in terms of the rate of convergence, which the authors stress throughout, but does not present either a major advance over [2]. It also does not seem to be a particularly practical method, due to the number of parameters needed; that is: while Figures 1 and 2 present very nice results, esp. on the synthetic dataset, it is very hard to know how much effort has gone into tuning those parameters on the instances or whether one could use one set parameters across instances of many different sizes. [1] Ohad Shamir. A stochastic PCA and SVD algorithm with an exponential convergence rate. In 303 Proceedings of the 32nd International Conference on Machine Learning, ICML 2015, Lille, France, 304 6-11 July 2015, pages 144–152, 2015. [2] Jialei Wang, Weiran Wang, Dan Garber, and Nathan Srebro. Efficient coordinate-wise leading 309 eigenvector computation. Journal of Machine Learning Research (2017) 1-25.

Reviewer 3



This paper proposes an algorithm which combines shift-and-invert preconditioning and gradient descent to compute leading eigenvector. Shift-and-invert preconditioning is a common preconditioning technique, and gradient descent is one of the most popular optimization algorithms, so this combination seems reasonable. They establish the complexity of using this method, by using a somewhat new potential function. Pros: The main difference of this paper with previous works is that it allows the multiplicity of the leading eigenvalue to be larger than 1, while most works assume an eigen-gap between the largest and second largest eigenvalue. In addition, this paper introduces a somewhat new potential function, that depends on the angle between a vector and the subspace spanned by the eigenvectors. More specifically, the proof uses two potential functions, one is log(u) and another is 1 – sin^2(v), where u and v depend on the angle. The second one is not new, but the first one log(.) seems to be new. Cons: There are quite a few issues to be addressed/clarified. 1. Complexity. Perhaps the biggest issue is: the complexity proved in this work is not shown to be better than the other existing results (a bit subtle; see explanation below). In fact, we need to discuss two cases: Case 1: the multiplicity of the largest eigenvalue is 1, i.e $\lambda_1 > \lambda_2 = \lambda_{p+1}$. In this case, the complexity of this work is not better than those in [Musco and Musco, 2015] and [Garber et al., 2016] (in fact, the same). Case 2: the multiplicity of the leading eigenvalue is larger than 1. In this case, the proposed rate is the “best“ so far. However, most other works do not consider this scenario, so this is not a fair comparison. Overall, the improvement only in Case 2 seems to be a minor progress. BTW: in Table 1, it seems that $\sqrt{\lambda_1}$ is missing from the numerator of the rate for this work''. 2. Limited Novelty of Proposed Algorithm There are already a few works which combine shift-and-invert preconditioning with some common base methods, such as accelerated gradient descent [Wang et al., 2017], accelerated randomized coordinate descent methods [Wang et al., 2017] or SVRG [Garber et al., 2016]. It is not surprising to use gradient descent (GD) for subproblems. From the arguments made in paper, I fail to see why GD will be a better choice. First, the authors claim that GD is better because Lanczos algorithms does not work for positive definite matrix, but GD does (line 39). However, some other base methods, such ascoordinate-wise'' methods [Wang et al., 2017], also work for positive definite matrices. Second, the authors claim GD will not be stuck at saddle points with probability one from a random starting point. The authors make this point by citing [Pitaval et al., 2015] (line 42). However, [Pitaval et al., 2015] is about a quite different problem. In addition, people tend to believe all reasonable first order methods will not be stuck at saddle points. The warm-start technique (line 158) is also not new to the community. The same formulation of warm-start has been used in [Garber et al., 2016]. 3 Algorithm Details The first negative sign in line 4 of Algorithm 1 should be removed. From equation (3), Riemannian gradient is equal to Euclidean gradient left multiplied by $(I - x_{t-1} x_{t-1}^T)$. In Algorithm 1, $y_t$ is the approximate Euclidean gradient. Therefore, the approximation Riemannian gradient in line 4 should be $(I - x_{t-1} x_{t-1}^T) y_t$. Another thing that requires clarification is the choice of stepsize. In Algorithm 1, authors pick the stepsize at every iteration such that the stepsize satisfies the second inequality in line 176. Why do the authors pick the stepsize at each iteration $t$, while the second inequality in line 176 is not dependent on each iteration $t$. Why don't we pick a constant stepsize before running the iterations? 4. Experiments The authors only compare Rimennian gradient descent solver with/without shift-and-invert preconditioning, and shift-and-inverted power method. As I mentioned before, the authors does not show the complexity of this work is better than [Musco and Musco, 2015] (Block Krylov) and [Garber et al., 2016] (shift-and-inverted power method). Therefore, it is reasonable to also consider comparing with these methods. 5. Minor issues: --It seems that the authors did not explicitly define Riemannian gradient. -- line 171 should be Lemma 3.2 rather than 3.3. After the rebuttal: I've increased my score by one, due to the good empirical performance. The tone of this paper is like "our contribution is to combine two methods, and get the best rate so far", and my above comments said, "combining two methods is rather straightforward, and the rate does not improve previous ones" --thus a clear reject. If the authors motivate the paper from an empirical perspective by saying "we find the combination of SI+GD works really well, so to promote it we provide theoretical justification", the contribution may be more clear. Considering the above reasons, I think this paper is slightly below the acceptance threshold. [Garber et al., 2016] Dan Garber, Elad Hazan, Chi Jin, Sham M. Kakade, Cameron Musco, Praneeth Netrapalli, and Aaron Sidford. Faster eigenvector computation via shift-and-invert preconditioning. In International Conference on Machine Learning, pages 2626–2634, 2016. [Musco and Musco, 2015] Cameron Musco and Christopher Musco. Randomized block krylov methods for stronger and faster approximate singular value decomposition. In Advances in Neural Information Processing Systems, pages 1396–1404, 2015. [Pitaval et al., 2015] Renaud-Alexandre Pitaval, Wei Dai, and Olav Tirkkonen. Convergence of gradient descent for low-rank matrix approximation. IEEE Trans. Information Theory, 61(8):4451–4457, 2015. [Wang et al., 2017] Jialei Wang, Weiran Wang, Dan Garber, and Nathan Srebro. Efficient coordinate-wise leading eigenvector computation. CoRR, abs/1702.07834, 2017. URL http://arxiv.org/abs/1702. 07834.